# Monolithic Side-Pumped Amplifier Based on an Yb-Doped Tapered-Fiber and Yielding 0.53 MW 9.3 ps Pulses

**Konstantin Bobkov [1,\*], Andrey Levchenko [1], Denis Lipatov [2], Alexey Guryanov [2], Mikhail Bubnov [1] and Mikhail Likhachev [1]**

1   Prokhorov General Physics Institute of the Russian Academy of Sciences, Dianov Fiber Optics Research Center of the Russian Academy of Sciences, 119991 Moscow, Russia
2   G.G. Devyatykh Institute of Chemistry of High Purity Substances of the Russian Academy of Sciences, 603950 Nizhny Novgorod, Russia
\*   Correspondence: wittkoss@gmail.com

**Abstract:** We demonstrated a simple design of a monolithic all-fiber side-coupled combiner for counter-pumped amplifiers that requires no special fiber processing systems for fabrication. The combiner based on a Yb-doped polarization-maintained tapered fiber with an output core diameter of 40 μm and a total length of 1.8 m exhibiting over 60% coupling efficiency of 976 nm 0.10 NA pump power was demonstrated and utilized to amplify 1064 nm 9.3 ps 1.84 MHz pulses up to 9.1 W of average power and 0.53 MW of peak power with near diffraction-limited beam quality. The demonstrated approach seems promising for further power scaling, retaining good output beam characteristics via design optimization.

**Keywords:** monolithic all-fiber system; Yb-doped tapered fiber; high peak power





## 1. Introduction

Ultrafast fiber lasers and amplifiers delivering high peak and average power in femtosecond and picosecond pulses with a diffraction-limited beam quality become an indispensable tool for many applications, first of all, micromachining. In [1], an all-fiber laser based on cascaded Mamyshev regeneration emitting ~1 MW peak power pulses with a duration of 40 fs was demonstrated, however, the pulse energy was only 50 nJ. Utilization of specially designed large mode area (LMA) fibers becomes vital when one needs to achieve high average power, high threshold of nonlinear effects, and high pump-to-signal conversion efficiency. Many good results in this regard were achieved using rod-type photonic crystal fibers (PCF), where MW-peak power level and μJ to mJ energy level were achieved directly after the fiber amplifier [2,3]. A disadvantage of PCF lasers is the necessity of utilizing a large number of bulk elements for coupling in and out signal/pump radiation, which results in loss of laser reliability. Significant progress in the achievement of LMA fiber laser reliability was made by developing fibers that could be spliced with preamplifiers. Splicing of PCF fiber becomes possible only with reduced core diameter, which allowed authors [4] to achieve ~70 kW peak power just after the active fiber. The best results in terms of simultaneous achievement of high peak and average power were demonstrated using tapered fibers, where a peak power of 0.7–2 MW was achieved [5–8]. In this case, pump is still coupled to the active fiber using bulk optics, but due to a large first clad diameter (200–350 μm), such lasers have a moderate sensitivity to misalignment, and achievement of MW peak power is possible together with a 150–200 W average power level.

The best results in terms of long-term reliability could be achieved only in a completely monolithic laser scheme, where both pump and signal radiation are coupled to an active fiber using a pump and signal combiner. The highest reported peak power of 1.3 MW for a monolithic all-fiber amplifier was reported in [9], however, the achievement becomes possible only by utilization of a co-propagating pump and signal amplifier scheme with a

Yb-doped tapered fiber having very large input core diameter of 36 μm and core numerical aperture (NA) of 0.064. It is well-known that it is nearly impossible to excite only the fundamental mode in step-index fiber with such a large core diameter (estimated cut-off wavelength of the first high order mode exceeds 3 μm in this case). Authors report a reasonably low $M^2$ of 1.19, but operation of the fiber in a few-modes propagation regime becomes obvious from the mode's image where the second side lobe is observed. Few modes propagation regime typically results in a deterioration of an output pulse shape due to the intermodal dispersion in an active fiber. In [9], there is a large pedestal in autocorrelation trace of the pulses, which we attribute to the presence of high-order modes (HOM). Similar amplifier design, but with a perfectly single-mode propagation regime, was demonstrated by our group in [10]. In this case, an input core diameter of an LMA tapered fiber was reduced down to 14 μm, which allowed us to achieve the first HOM cut-off of 0.95 μm relative to a Ge-doped pedestal surrounding the core. Very high $Yb^{3+}$ concentration in the core enabled us to reduce the fiber length down to 21 cm. It allowed us to keep a reasonably high threshold of nonlinear effects—chirped pulses being amplified up to 350 kW of peak power were still compressible down to sub-ps duration. It seems to be a limit to this technique, as increasing the core diameter and retaining its single-modeness is hardly possible as well as it is nearly impossible to further decrease the fiber length.

It should be noted that amplifier configuration with a co-propagating pump and signal is inferior in terms of peak power to an amplifier with a counter-propagating pump and signal. The reason for this is that a signal power in a co-pumped scheme rapidly develops and then stays still or ever reabsorbs and accumulates nonlinear effects, while in a counter-pumped scheme, it gradually increases in power, and thus accumulates a lower level of nonlinear effects because of a lower nonlinear length. Indeed, all the record results in terms of peak power for usual, non-completely monolithic amplifiers were obtained with a counter-propagating pump and signal. However, application of counter-pumping for a monolithic amplifier scheme is a challenge. Indeed, standard pump combiners have a passive fiber core diameter of less than 30 μm. Moreover, nonlinearity accumulated in the pump combiners built of passive fiber will strongly limit maximum peak power at a level far below MW. In the current work, we propose and realize a novel approach for building a completely monolithic single-mode amplifier delivering sub-MW peak power pulses. The main idea is based on the non-fusion fixing of a pump-feeding fiber on a side surface of an active tapered fiber just near its output thick end. The highest peak power of 0.53 MW was achieved in a completely monolithic amplifier operated in the single-mode regime.

## 2. Overview of Existing Pump Combiners Design

Monolithic all-fiber systems are created with pump and signal combiners of two types, which differ in a pump radiation coupling way: Through a delivery fiber butt-end or through its side surface. The design of a pump combiner of the first type is a fiber bundle, consisting of a signal fiber surrounded by a few pump fibers, tapered down by means of complex fiber processing systems to the outer diameter of a delivery fiber and spliced to it [11]. The tapering process leads to a mismatch in diameters of the signal fiber core and the delivery fiber core, this, in the case of LMA fibers, leads to a signal coupling into high-order modes in the delivery fiber core and onset of the mode instability [12] and the long-term mode degradation [13] effects. Moreover, a combiner of such a design is not suitable for the implementation of counter-pumped scheme highly desirably to obtain high peak power because a signal power will couple into pump-feeding ports and harm pump diodes. Relatively recently, it was proposed to taper the bundle without the signal fiber, which was then inserted in the tapered fiber bundle (TFB) after chemical etching of its cladding down to a diameter of a hole in the TFB that then precisely spliced to the delivery fiber with active alignment process to avoid high-order modes excitation [14]. Such a design allowed authors of [15] to create a counter-pumped combiner with a delivering double-clad fiber with core/cladding diameters and numerical aperture (NA) of 25/400 μm and 0.065/0.46, correspondingly. The $M^2$ factor of a counter-pumped system based on the combiner was

1.364 at 15.9 W of signal and 1.498 at 1067 W. However, it is apparent that it will be extremely difficult to avoid excitation of high-order modes by using a signal fiber with larger core diameter, and also, at some point, one will be unable to use polarization-maintaining fibers of PANDA type due to etching its stress rods located in its first reflective cladding.

In the second type combiners, a pump radiation coupled from a pump-feed fiber into a delivery fiber reflective cladding through their side surfaces, which are in close contact along the length of the fibers, due to violation of the total internal reflection (TIR) condition. The simplest example of such a design is the so-called GTWave fiber which consists of one active fiber and a few pump-feed fibers placed in close contact and covered by a low-index polymer coating [16]. The maximum achievable coupling efficiency of the GTWave fiber could be estimated as $D_{in}^2/(D_{in}^2 + D_{out}^2)$: Putting $D_{in}$ = 400 μm and $D_{out}$ = 105 μm, we obtain 93% coupling efficiency. However, the efficiency only tends to that value at a few meters long interaction length, while it comes in confrontation with demand to shorten an active fiber length in order to decrease nonlinear effect impact on an amplified signal. Moreover, the production of the GTWave fiber and, for example, LMA tapered fiber separately is a non-trivial task, while producing the GTWave fiber based on an active tapered fiber is an extremely complicated task.

It is possible to achieve almost 100% coupling efficiency with only a few centimeters of interaction length by stimulating violation of the TIR condition via tapering a pump-feeding fiber and fusing it into the reflective cladding of an active or a delivering fiber. In [17], a side-coupled fused pump combiner where two pump-feeding fibers with parameters 220/242 μm and 0.22 NA were tapered down to an outer diameter of ~30 μm at length of ~1 cm and then, using special fiber processing system, side surface fusion spliced at depth of ~10 μm along the whole length of the tapered part into a delivering fiber with parameters 20/400 μm 0.06/0.46 NA. The combiner had a coupling efficiency of ~97% at 976 nm wavelength and was used to develop counter-pumped amplifier with output beam $M^2$ factor of 1.05 at coupled pump power of 3070 W. In [18], a similar side-coupled fused combiner based on a 40/400 μm 0.06/0.46 NA delivering fiber and exhibiting 98.1% coupling efficiency carrying 2882 W pump power was demonstrated, however, no information on the output beam quality of the combiner was provided.

Apparently, the side surface fusion splicing process inevitably leads to a formation of micro and macro bends in an LMA delivery fiber, which potentially distorts polarization-maintaining properties of the fiber and results in high-order modes excitation. In this paper, we demonstrate a simple side-coupled monolithic pump combiner design hybriding ideas of the GTWave fiber and a tapered side-fused combiner, and requiring no special fiber processing system for its fabrication. The pump combiner of our design was realized using an all-glass polarization-maintaining Yb-doped tapered fiber with a core/first cladding/second cladding diameters and numerical aperture of 40/320/370 μm and 0.1/0.28, correspondingly, and three pump-feeding all-glass 105/125 μm 0.22 NA fibers. The coupling efficiency for multimode radiation at 976 nm with 0.10 NA was estimated to be over 50%. Using the combiner, an all-fiber monolithic master oscillator power amplifier was developed. It amplified pulses at 1064 nm with duration of 9.3 ps, average power of 50 mW and repetition rate of 1.84 MHz up to 9.1 W of average power, that corresponding to peak power of 0.53 MW.

## 3. Theory

The idea of our design, depicted in Figure 1, is quite simple: A pump-feeding all-glass 105/125 μm 0.22 NA fiber is bitapered and fixed by an ultraviolet-cured low-index polymer at LMA fiber (in our case, it was in-home made Yb-doped tapered fiber with diameters for core/first reflective cladding/second reflective cladding of an output end, where the combiner is to be developed, of 40/320/370 μm) with a removed protective polymer coating and, in the case of presence, a second reflective cladding, ensuring a good contact between a down-taper and waist regions of the pump-feeding fiber and the first reflective cladding of the LMA fiber. The down-taper region forces TIR condition violation and coupling of a

high and a medium NA radiation into the first reflective cladding of the LMA fiber, with further partial loss of high NA radiation, while a low NA radiation almost completely passes into an up-taper region of the pump-feeding fiber through the waist region (we schematically showed two first coupled rays for all the NAs). The residual pump radiation which undergoes an increase of NA can be used once again by fusion splicing the up-taper end of the pump-feeding fiber to another pump-feeding fiber down-taper end.

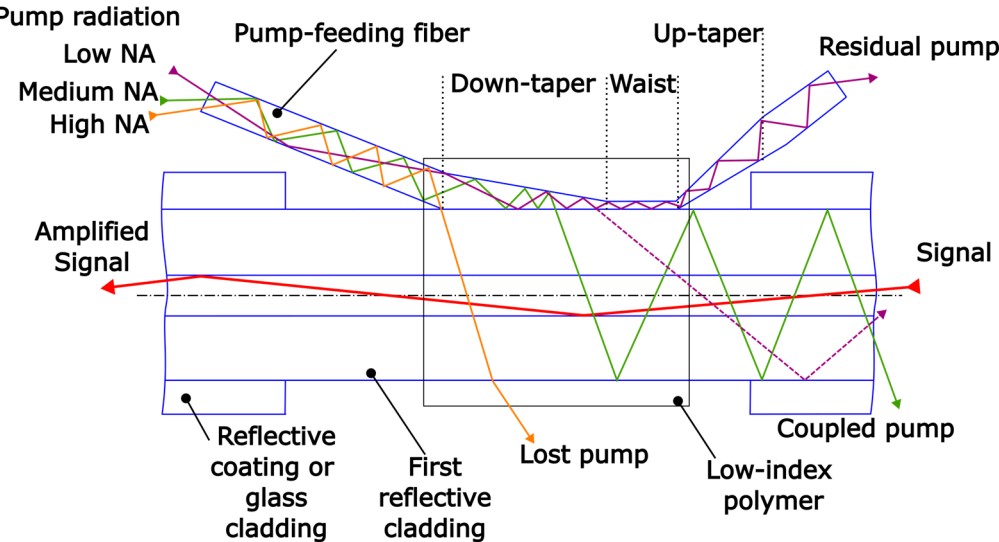

**Figure 1.** The scheme of the side-coupled pump combiner. Protective polymer coating of the pump-feeding fiber and the LMA fiber were not shown.

Among the existing LMA fiber designs, active tapered fibers have an undisputed advantage—their signal input end, usually having core/cladding diameters of around 10/125 µm, is easily fusion spliced with conventional fibers with a 125 µm outer diameter, enabling the creation of all-fiber amplifiers with perfect output beam quality due to adiabatic propagation of the fundamental mode from thin to thick end of the tapered fiber. Our laboratory develops polarization-maintaining tapered fibers with high $Yb^{3+}$ ions concentration, an output core diameter up to 50 µm, all-glass construction (the first reflective silica cladding is surrounded by a second fluorine-doped silica cladding), and a relatively short length of 1–2 m. The all-glass design exhibits higher temperature capabilities in comparison to fibers with reflective polymer cladding, however, in this case, the numerical aperture of the first reflective cladding is only 0.28, and thus the down-tapered part of the pump-feeding fiber in the side-coupled combiner should be considerably longer than in the case of side-coupled fused pump combiners based on a fiber with a reflective polymer with 0.46 NA.

To find an optical length of the down-taper part of the pump-feeding fiber, we conducted simulations in the ray optics Zemax software, where the model based on Figure 1 was developed in 3D and investigated. The pump-feeding fibers support propagation of an extremely high number of transverse modes, and propagation simulation will be very time-consuming. For this reason, it is more convenient to use the ray optics approach. The pump-feeding fiber had parameters as follows: A core refractive index and diameter are 1.4507 and 105 µm, the down-tapered part length varied from 10 to 80 mm, the waist diameter and length were 30 µm and 20 mm, correspondingly. The waist parameters were chosen from considerations of easiest fiber handling (thinner fibers are too fragile) and fiber fixing at the delivery fiber side surface. The delivery fiber had a cladding diameter and refractive index of 300 µm and 1.4507, correspondingly, the fluorine-doped cladding refractive index was 1.4234. The fixing polymer coating refractive index was 1.372. The simulations were conducted for radiation with 976 nm wavelength and NAs of 0.10, 0.15, 0.20.

The simulation results show that the coupling efficiency curves for all the NAs almost saturate at a length of ~50 mm, and the highest efficiency of ~65% is predicted for 0.10 NA radiation (see Figure 2a). The efficiency is slightly higher at a length of 70 mm in comparison to a length of 50 mm, but that negligible increase costs an additional 20 mm of fiber etching, resulting in higher complexity of the combiner production. It is worth noting that because there is no fusion of the pump-feeding fiber into the delivery fiber, their side surfaces are only in close contact, the coupled radiation exhibits a donut-like shape intensity distribution, i.e., preferably high-order modes in the delivery fiber are excited. Our simulation also shows that only a small fraction of the pump radiation passes the waist region of the bitapered fiber (~2% for 50 mm down-taper length and 0.10 NA), see Figure 2b. Therefore, for the optimal case of ~50 mm down-taper length and the pump radiation with 0.10 NA, about 35% of the pump power will be lost.

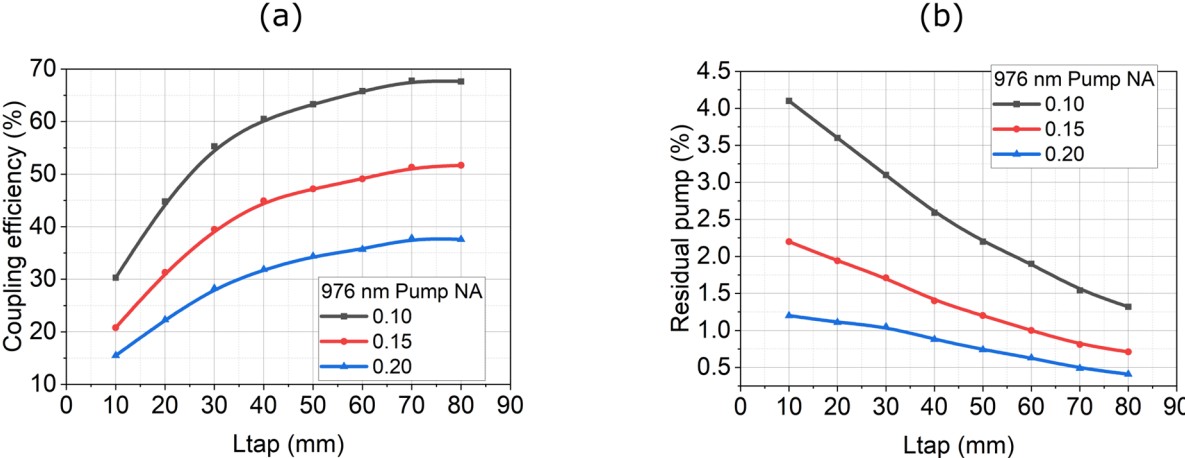

**Figure 2.** (**a**) Simulated pump coupling efficiency of the proposed pump combiner design for three pump radiation NAs at 976 nm. (**b**) Simulated residual pump fraction of the combiner for three pump radiation NAs at 976 nm.

## 4. Developed Side-Coupled Pump Combiner

In accordance with the obtained simulation results, we developed a counter-pumped side-coupled combiner with three bitapered pump-feeding fibers fixed on a side surface of a Yb-doped tapered fiber. We used a standard Yb-doped fiber that our group produces with parameters: Core/first silica cladding/second F-doped silica cladding diameters and NAs of 40/320/370 µm and 0.1/0.28, correspondingly, and a total length of 1.8 m (see Figure 3a). The second F-doped silica cladding was etched in the hydrofluoric acid at a position starting at ~40 mm from the fiber thick output end and at a length of ~70 mm. The starting position was chosen keeping in mind that we have to keep some length of the active fiber between its output end and the etched region just to have an opportunity to fix the device in some housing. The etching was performed by placing a part of the fiber into a reactor with dimensions of 70 × 2 × 2 mm filled with hydrofluoric acid. The pump-feeding fibers were bitapered from a conventional 105/125 µm 0.22 NA fiber by chemical etching as well. The U-bended fiber was pulled from a container with the hydrofluoric acid in such a way to achieve desired geometrical parameters by means of a motorized stage controlled by a personal computer. The tapered regions length of ~50 mm, the waist length of ~20 mm and diameter of ~30 µm (see Figure 3b). The accuracy of the etching was at 10% level caused by the degradation of the acid.

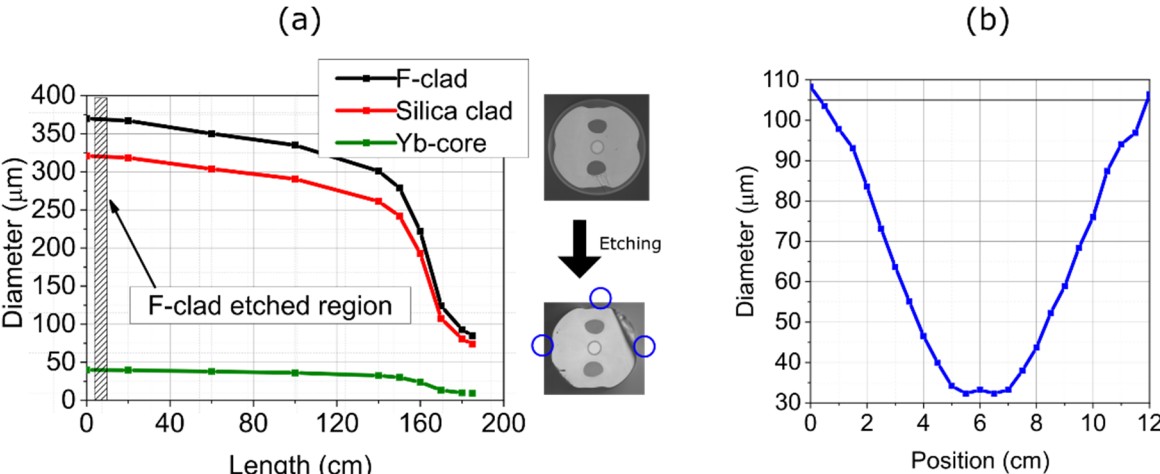

**Figure 3.** (**a**) Used Yb-doped tapered fiber diameters on length dependence and cross-section photo before and after etching of its second F-doped silica cladding; blue circles on cross-section photo schematically depict positions of the pump-feed bitapered fibers. (**b**) Typical diameter dependence on the length of the etched bitapered fiber.

The bitapered fibers were fixed along the Yb-doped tapered fiber in grooves at its first silica cladding, created on purpose for better mixing of pump radiation, assuming higher overlapping of side surfaces. The fixing was made by an ultraviolet-cured polymer with a refractive index of 1.372 at 976 nm. The developed device with coupled 976 nm and 0.10 NA pump radiation is shown in Figure 4: From the visible greenish cooperative luminescence of trivalent ytterbium ions, we can conclude that the pump radiation coupled into the tapered fiber, not from the beginning of the down-taper region, but at a position of ~30 mm. The device should be packaged into a rigid housing, in which it has to be fixed in the same manner as it is shown in Figure 4.

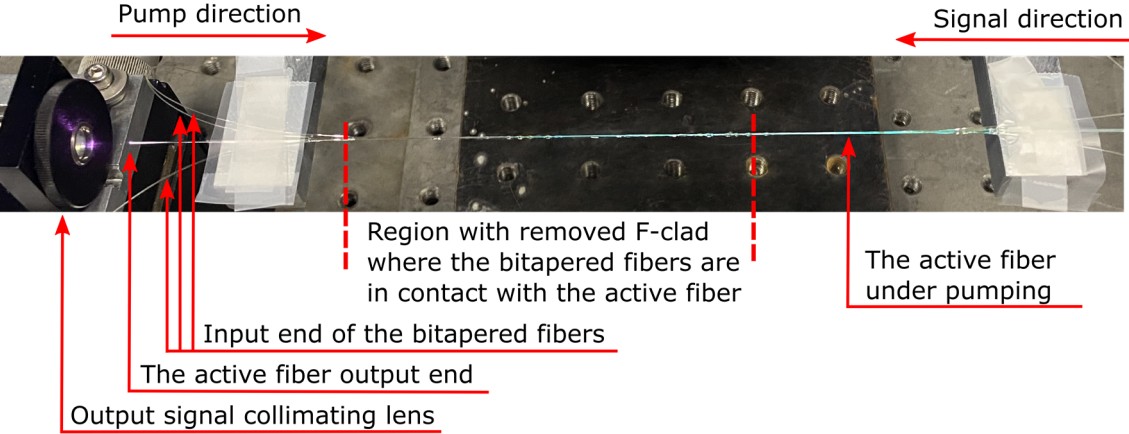

**Figure 4.** The developed side-pumped combiner on the Yb-doped tapered fiber under pumping with 976 nm 0.10 NA radiation.

## 5. Experiments

We implemented a simple master oscillator power amplifier (MOPA) setup, consisting of a home-made semiconductor saturable absorber mirror (SESAM) mode-locked fiber laser generating pulses centered at 1064 nm with a repetition rate of 18.4 MHz, two low-power amplification stages with acousto-optic modulator in between and the developed side-coupled monolithic combiner in a final amplification stage (see Figure 5). The AOM was utilized to reduce the signal repetition rate to 1.84 MHz in order to increase pulse energy at the system output. The tapered fiber was seeded with pulses with 50 mW average

power, 9.3 ps duration, and a repetition rate of 1.84 MHz. All the parts of the MOPA were based on homemade 6/125 μm polarization-maintaining PANDA-type fibers. To pump the fiber, two pigtailed multimode diodes stabilized at the wavelength of 976 ± 0.5 nm with 0.10 NA and output power of ~15 W were fusion spliced to two pump-feeding ports of the combiner, while the residual pump from its up-tapered ends was coupled into a conventional 2-to-1 pump combiner, which output end was fusion spliced to the third pump-feeding fiber of our side-combiner. It is worth noting that prior to the realization of the side-coupling combiner, the utilized Yb-doped tapered fiber was also investigated in the same MOPA, but with bulk optics pump coupling in order to study the combiner impact on the output radiation characteristics of the tapered fiber, such as polarization extinction ratio and beam quality, and to estimate its coupling efficiency.

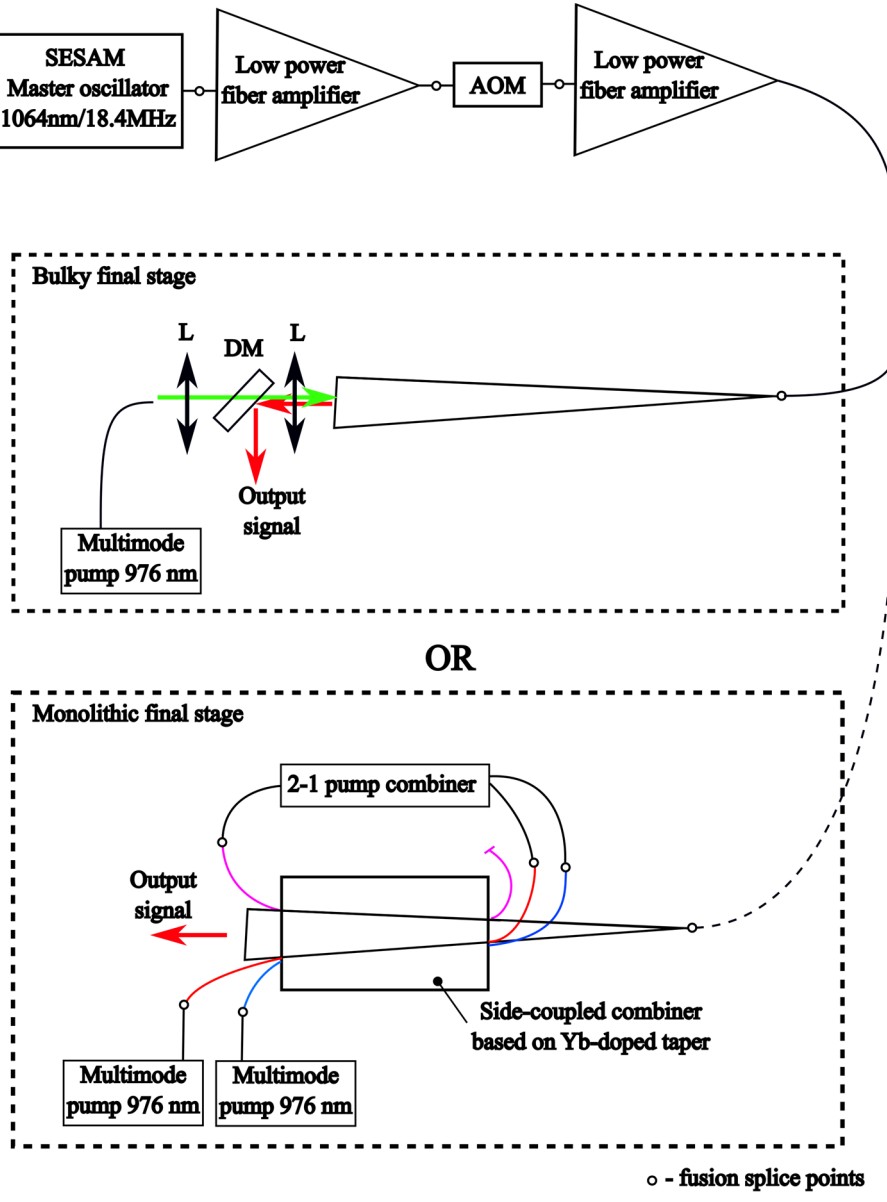

**Figure 5.** The realized all-fiber MOPA, both bulky and monolithic final stages are shown. Low-power pulses were generated in SESAM master oscillator, amplified at mW level in the low-power fiber amplifiers, and then amplified at 9.1 W level in the bulky or in the monolithic final stage. The green arrow represents the multimode pump radiation, and the red arrow represents the output signal radiation.

Figure 6 shows how the output power of the amplified signal at 1064 nm evolves with the pump power increase. For the bulk optics scheme, we assume that pump power coupling efficiency is close to 100% because the pump radiation coupled from a 200/220 μm pump combiner fiber to the first reflective cladding of the tapered fiber with a diameter of 320 μm by means of two identical lenses. At the pump power level of 25 W, one can see the curve kinks due to the nonlinear effect onset. The side-pumped curve for our monolithic combiner consists of two linear regions with a breaking point near 25 W of pump power. From the bulk and monolithic curve points with the same signal power, we can estimate coupling efficiency for our monolithic design: It started from 50% at low pump power level and reached 90% at maximum pump power. We omitted the first few points because the signal power was too low there for correct estimations. The observed increase in the coupling efficiency at the second region of the monolithic curve, as well as the two-piece appearance of the curve, is most likely because of some deformation of the Yb-doped tapered fiber and the pump-feeding fibers. This suggestion is supported by the observation of the beam quality factor $M^2$ and the polarization extinction ratio. It can be seen from Figure 7a, that the $M^2$ factor curve of the side-combiner also shows two-pieces behavior, exactly copying its pump-to-signal curve: At first, it stays at 1.15 level up to pump power of 17.5 W, but then, it increases up to 1.29 at pump power of 27.2 W, while for the bulk optics scheme, $M^2$ factor value lays below 1.2. The polarization extinction ratio (PER) measurements showed no considerable difference between the schemes, however, some tendency of PER decrease in the side-coupled scheme was observed (see Figure 7b). Thus, we suggest that heating the fiber bungle at the position of optical contact between fibers may noticeably change the coupling efficiency but also disturb the beam quality. Finally, at the average power of 9.1 W, the side-pumped scheme exhibits a very low level of stimulated Raman scattering, whose integral part is lower than 0.02% (see Figure 8a). The autocorrelation function of the obtained pulses exhibits a perfect Gaussian shape with duration of 9.3 ps (see Figure 8b), so the peak power was estimated to be 0.53 MW. The output power stability was at a level of 1% at 9.1 W for a time interval of 1 h.

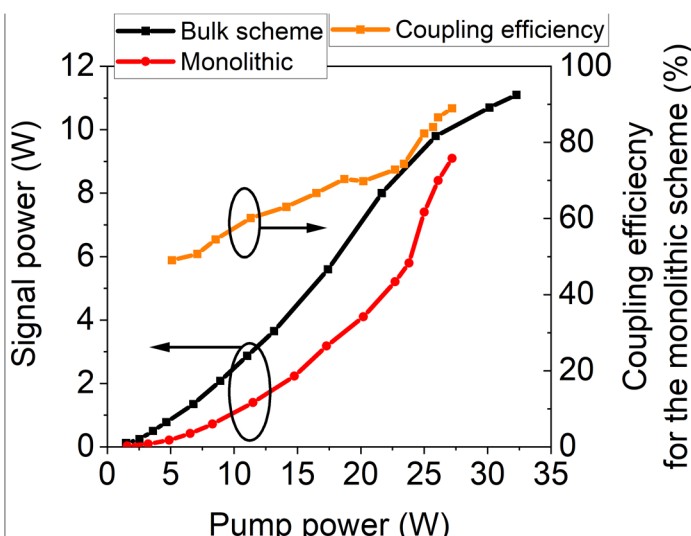

**Figure 6.** Dependence of signal power on pump power for bulk optics and side-coupled monolithic schemes and coupling efficiency for the monolithic scheme.

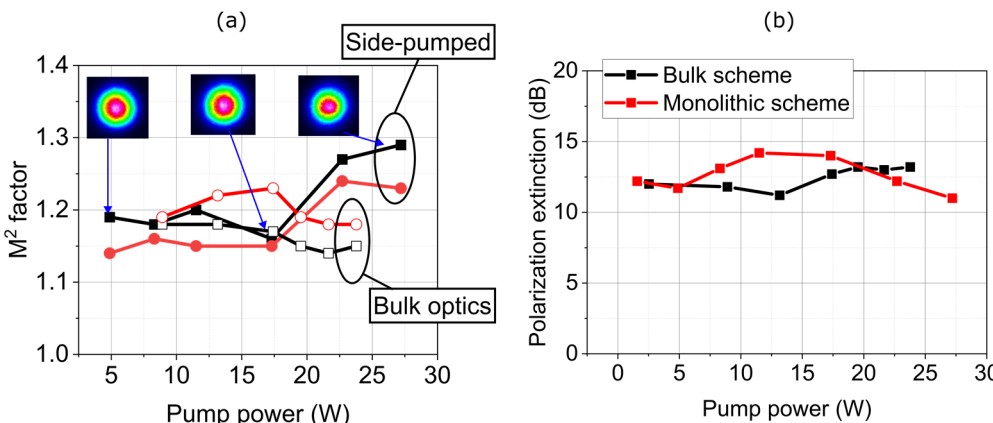

**Figure 7.** (**a**) The M$^2$ factor dependence on the pump power. The red curves are for "X" axis and the black curves are for "Y" axis of the M$^2$ factor. (**b**) The polarization extinction ratio evolution in both schemes.

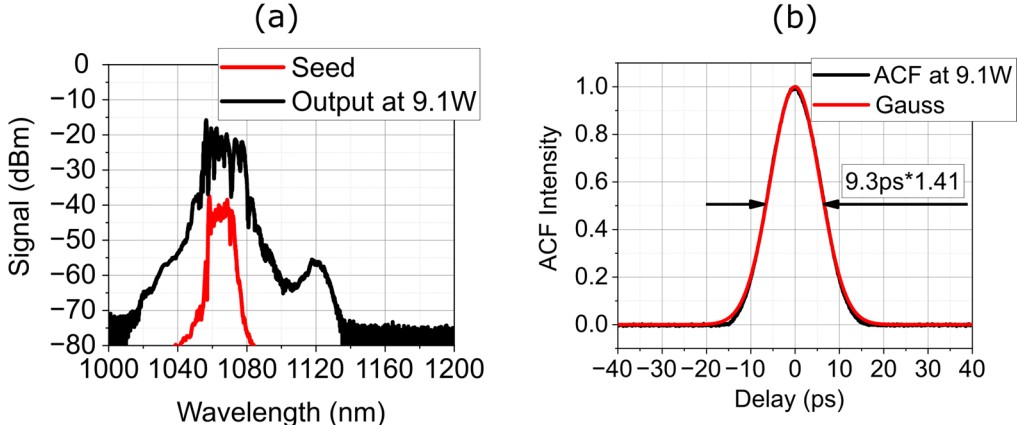

**Figure 8.** (**a**) The seed and output signal spectra of our monolithic side-pumped scheme. (**b**) The autocorrelation trace of amplified pulses.

Further pump power increase leads to burning of the side-coupled pump combiner. We suggest that it is due to strong scattering at the surface of pump-feeding fibers that caused heating of the pump combiner. This scattering was caused by surface roughness which appears during the fiber etching process. It is interesting to note that attempts to fuse pump-feeding fiber into the Yb-doped tapered fiber result in an increase of pump coupling efficiency and simultaneously strong mode shape deterioration due to mode coupling in the core of tapered fiber as a result of incorporation of stresses. Therefore, further progress in power scaling of the device is possible by applying another production process of pump-feeding fiber tapers, for example, using heating and pulling. Moreover, we believe that utilization of a special fiber processing system might allow one to fuse pump-feeding fiber with the Yb-doped tapered fiber without significant mode shape deterioration.

It also should be noted that the developed device can amplify any radiation within ytterbium ions amplification band, not only 1064 nm pulses with ps duration.

## 6. Conclusions

We demonstrated a novel and simple to implement and demanding no special fiber processing system design of a side-pumped monolithic combiner based on a Yb-doped polarization-maintaining fiber with output core diameter of 40 μm. The combiner coupling efficiency exceeded 50% for 976 nm 0.10 NA multimode pump radiation and reached nearly 90% for pump power in excess of 25 W. We have demonstrated, for the first time, the possibility of building a counter-pumped monolithic tapered fiber amplifier with sub-MW

output peak power. To the best of our knowledge, it is the highest peak power achieved in monolithic fiber systems with diffraction-limited beam quality and a perfect, pedestal-free ps pulse shape.

**Author Contributions:** K.B. worked on the design, did the simulations, fabrication and measurements of the devices and wrote the whole paper; A.L. fabricated the active tapered fiber; D.L. and A.G. produced the fibers preforms; M.B. and M.L. supervised the project. All authors have read and agreed to the published version of the manuscript.

**Funding:** This work was supported by the Center of Excellence "Center of Photonics" funded by the Ministry of Science and Higher Education of the Russian Federation under Contract 075-15-2022-315/22.

**Institutional Review Board Statement:** Not applicable.

**Conflicts of Interest:** The authors declare no conflict of interest.

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
