# Peer review of "Monolithic Side-Pumped Amplifier Based on an Yb-Doped Tapered-Fiber and Yielding 0.53 MW 9.3 ps Pulses"

_photonics, doi:10.3390/photonics9100771_

Round 1

Reviewer 1 Report

The authors demonstrated a simple design of a monolithic all-fiber side-coupled combiner for counter-pumped amplifiers that requires no special fiber processing systems for fabrication. This is a hot topic in the high-power laser research field, which has much beneficial for boosting the fiber laser power. However, in my opinion, the paper cannot be published based on the present version, because in which the manuscript structure and some mistakes or problems should be improved or revised first. My comments are listed as below:

1.       The last paragraph in Introduction: “It should be noted that amplifiers configuration with co-propagating pump and 63 signal is inferior in terms of peak power to an amplifier with counter-propagating pump 64 and signal”. The authors should demonstrate the reason for this.

2.       The manuscript structure should be improved. For example, part 2(overview of the existing pump combiners design) should be added into the Introduction. In addition, some references should be cited in the first sentence in the Introduction to prove it, such as DOI: 10.1016/j.mtphys.2022.100622, and DOI: 10.1364/OPTICA.4.000649.

3.       The model number of the two fibers in Figure 1 should be presented in the paper, which is an important information to the readers. In addition, the input signal also should be plotted in this picture.

4.        The authors said that the coupling efficiency curves for all the NAs saturates at 50 mm position in picture2a, and the highest efficiency is  65% for 0,10 NA. But it seems that the highest coupling efficiency is more than 65% at length of ~70 mm.

5.       What is the model number of the polarization maintaining fiber used in the MOPA? In addition, why a AOM is inserted between the two low power fiber amplifiers?

6.       The authors should check the manuscript carefully. For example, in the first paragraph in part 5 Experiments, the repetition rate of the seed source is 1.84 MHz or 18.4 MHz? A space should be inserted between 0.10 and NA in the abstract.

7.       What is the full name of the “MM pump” in figure 5? Is it multimode? The author should present it in this picture.

8.       The English and grammar should be polished further.

Author Response

  1. The last paragraph in Introduction: “It should be noted that amplifiers configuration with co-propagating pump and 63 signal is inferior in terms of peak power to an amplifier with counter-propagating pump 64 and signal”. The authors should demonstrate the reason for this.

       The reason for this is that a signal power in a co-pumped scheme rapidly develops and then stays still or ever reabsorbs and accumulates nonlinear effects, while in a counter-pumped scheme it gradually increases in power, and thus accumulates lower level of nonlinear effects because of a lower nonlinear length. – added at 68.

  1. The manuscript structure should be improved. For example, part 2(overview of the existing pump combiners design) should be added into the Introduction. In addition, some references should be cited in the first sentence in the Introduction to prove it, such as DOI: 10.1016/j.mtphys.2022.100622, and DOI: 10.1364/OPTICA.4.000649.

      Merging of the introduction and the overview parts will result in an extremely hard to read and understand introduction. We cited a paper devoted to Mamyshev regenerator based laser in the introduction at 24.

  1. The model number of the two fibers in Figure 1 should be presented in the paper, which is an important information to the readers. In addition, the input signal also should be plotted in this picture.

(in our case it was in-home made Yb-doped taped fiber with diameters for core/first reflective cladding/second reflective cladding of an output end, where the combiner is to be developed, of 40/320/370 µm) – added at 149.

  1. The authors said that the coupling efficiency curves for all the NAs saturates at 50 mm position in picture2a, and the highest efficiency is  65% for 0,10 NA. But it seems that the highest coupling efficiency is more than 65% at length of ~70 mm.

      That is true, however the difference in coupling efficiency between 50 mm and 70 mm is negligible, while in the latter case the one should remove additional 20 mm of second reflective silica cladding from a delivery fiber by chemical etching that complicates production process. – added at 196.

  1. What is the model number of the polarization maintaining fiber used in the MOPA? In addition, why a AOM is inserted between the two low power fiber amplifiers?

      The fibers were in-home made 6/125 µm PANDA ones. – added at 254.

      The AOM was utilized to reduce signal repetition rate from 18.4 MHz to 1.84 MHz in order to increase pulses energy at the system output. – added at 251.

  1. The authors should check the manuscript carefully. For example, in the first paragraph in part 5 Experiments, the repetition rate of the seed source is 1.84 MHz or 18.4 MHz? A space should be inserted between 0.10 and NA in the abstract.

      The seed laser fundamental repetition rate was 18.4 MHz, and we used acousto-optic modulator to decrease pulses repetition rate down to 1.84 MHz in order to increase its energy.

      Placed space before all the “NA”s in the text.

  1. What is the full name of the “MM pump” in figure 5? Is it multimode? The author should present it in this picture.

      Added at Fig. 5.

  1. The English and grammar should be polished further.

       Corrected.

Reviewer 2 Report

The work "Monolithic side-pumped amplifier based on an Yb-doped tapered -fiber and yielding 0.53 MW 9.3 ps pulses" by the authors of Bobkow et. al. is interesting and should be published. However, it requires some supplements for a full understanding of the presented solution.

1.please explain how and with what accuracy the etching is performed

2.why the etching of PM fiber was chosen in the direction shown in figure 3,

3.figure 4 should be presented in more detail,

4. please present the experimental "bulk" scheme with which the results of the new proposed scheme are compared

Author Response

1.please explain how and with what accuracy the etching is performed

The etching of the active fiber was performed by placing a part of the fiber into a reactor with dimensions of 70x2x2 mm filled with the hydrofluoric acid. – added at 217.

The pump-feed fibers bended in U-letter style, placed into a container with the acid and pulled out by means of motorized linear stage – added at 224.

The accuracy of the etching was at level of about 10 %, occuring from degradation of the acid. – added at 226.

2.why the etching of PM fiber was chosen in the direction shown in figure 3,

We realized counter-pumped combiner, so in an ideal case we should place pump-feeding fiber just at the output end of the active fiber. The starting position for etching was chosen keeping in mind that we have to keep some length of the active fiber between its output end and the etched region just to have opportunity to fix the device in a some housing. – added at 217.

3.figure 4 should be presented in more detail,

Added a few more details on it.

4.please present the experimental "bulk" scheme with which the results of the new proposed scheme are compared

Showed bulky variant as well as monolithic at Figure 5.

Reviewer 3 Report

The manuscript by Konstantin Bobkov et al. describe a simple design of a monolithic all-fiber side-coupled combiner for counter-pumped amplifiers that requires no special fiber processing systems for fabrication. Basic simulations have been performed, and based on simulation results, monolithic all-fiber system have been designed, as well as basic characteristic parameters are obtained.

The manuscript contains interesting results, and can be accepted for publication in journal Photonics with major revisions.

Please find comments and questions below:

1.      Please explain all abbreviations, e.g. MOPA, SESAM, etc.

2.      Please check the whole manuscript for spelling errors, e.g. “acheaved” in line 28.

3.      The upper part of Figure 1 is cropped. Please improve this. In addition, does the pump fiber have any cladding?

4.      Line 171: Authors mention simulation using Zemax software. Have these simulation been done in 2D or 3D? Why the authors chose to do the simulations using ray optics module, and not wave optics modules?

5.      Please improve Figure 5, e.g. please show light propagation direction.

6.      Although Figure 6 shows that the coupling efficiency is below 1%, the manuscript text (line 255) contains the following statement: “50% at low pump power level and reaches 90% at maximum pump power”. Are these typos? Please improve Figure 6, so it is clearer for the readers to understand. E.g. the legend contain only two values, but there are three lines of different colour.

7.      Line 251: “The side-pumped curve for our monolithic combiner consists of two linear regions: the first one with ~34% slope and the second one with ~86% slope.”

1)      How could the reader understand the slope has indicated percentage?

2)      What are the physical reasons why there are two distinct regions with different slopes?

8.      Developed monolithic side-pumped amplifier based on an Yb-doped tapered-fiber is tested only for specific parameters (wavelength, repetition rate, and pulse duration). Does this system allow other types of input signal amplifications?

Author Response

  1. 1.Please explain all abbreviations, e.g. MOPA, SESAM, etc.

      Corrected.

  1. Please check the whole manuscript for spelling errors, e.g. “acheaved” in line 28.

           Corrected.

  1. The upper part of Figure 1 is cropped. Please improve this. In addition, does the pump fiber have any cladding?

      Corrected. The pump fiber has polymer cladding, that was not shown as well as polymer cladding the delivery fiber for the sake of simplicity of the figure. – added at 176.

  1. Line 171: Authors mention simulation using Zemax software. Have these simulation been done in 2D or 3D? Why the authors chose to do the simulations using ray optics module, and not wave optics modules?

      We have implemented the model using Zemax in 3D. The pump-feeding fibers support propagation of an extremely high number of transverse modes, which propagation simulation will be very time-consuming. For this reason it is more convenient to use ray optics approach. – added at 180     – added at 182.

  1. Please improve Figure 5, e.g. please show light propagation direction.

      An explanation of the scheme was added into its description.

  1. Although Figure 6 shows that the coupling efficiency is below 1%, the manuscript text (line 255) contains the following statement: “50% at low pump power level and reaches 90% at maximum pump power”. Are these typos? Please improve Figure 6, so it is clearer for the readers to understand. E.g. the legend contain only two values, but there are three lines of different colour.

      Corrected “coupling efficiency” at all the figures to %s.

  1. Line 251: “The side-pumped curve for our monolithic combiner consists of two linear regions: the first one with ~34% slope and the second one with ~86% slope.”

1)      How could the reader understand the slope has indicated percentage?

We decided to omit this information, because it has no practical use.

2)      What are the physical reasons why there are two distinct regions with different slopes?

      The observed … two-piece appearance of the curve, is most likely because of some deformation of the Yb-doped tapered fiber and the pump-feeding fibers. – added at 284.

  1. Developed monolithic side-pumped amplifier based on an Yb-doped tapered-fiber is tested only for specific parameters (wavelength, repetition rate, and pulse duration). Does this system allow other types of input signal amplifications?

      The device was also investigated for continuous-wave signal at 1064 nm with the same results in achieved average power. It wasn’t tested for other conditions (like 1030 nm signal or ns pulse duration), but there are no reasons to see any difficulties for the device to amplify any other signal wavelength within ytterbium ions amplification band. – added at 310.   

Reviewer 4 Report

The author's report on a monolithic all-fiber side-coupled combiner for counter-pumped amplifiers with a record highest peak power in sub-MW power range operated in single mode regime. The novel and simple design of the 40 µm in diameter (output core) fiber with Yb doping successfully maintains the polarization. The combiner coupling efficiency of about ~50 % for 976 nm with 0.10 NA multimode pump radiation makes it unique with the possibility of building counter-pumped monolithic tampered fiber amplifier with sub-MW output peak power. The authors successfully explored the idea based on non-fusion fixing of a pump-feeding fiber on a side surface of an active tapered fiber right next to its output thick end.

The authors present an in-depth and informative introduction with references to the relevant recent literature. The article contains important and significant novel results that will be of great interest for the broad scientific readership of the journal.

For the reasons mentioned above, I suggest this work deserves publication in photonics, provided the following (major) comments and suggestions are fulfilled:

1.       The y-axis in Figure 2a refers to Coupling Efficiency and not efficiency. The typo error needs to be corrected in Figure 2a. The legend of Figure 2a hides the data points corresponding to NA0.10. Consider moving the legend to the lower right corner for clearer display of the data points.

2.       Figure 3a caption (line 217, page 5) mentions red circles but there are not any of those in the figure. The authors referred to blue circles in the photo. The authors must add scale bars to the photographic images of the cross-section before and after etching process.

3.       The reference to Figure 7 made on page 7 (line 260) should be changed to Figure 7a.

4.       The authors successfully demonstrate the output signal spectra in Figure 8a. How does the output signal vary with time? The authors should provide the stability data for the output signal as a function of time.

5.       As shown in Figure 4, the fiber setup is placed horizontally straight for measurements. How would the output signal change if the fiber is tested under bending conditions? The authors should provide data to support the robustness of the fiber system.

6.       How does the output peak power depend on the diameter of the all-fiber side coupled combiner? What is the reason behind choosing specific size of 40 µm diameter for Yb-doped polarization maintained tapered fiber?

Author Response

  1. The y-axis in Figure 2a refers to Coupling Efficiency and not efficiency. The typo error needs to be corrected in Figure 2a. The legend of Figure 2a hides the data points corresponding to NA0.10. Consider moving the legend to the lower right corner for clearer display of the data points.

Corrected.

  1. Figure 3a caption (line 217, page 5) mentions red circles but there are not any of those in the figure. The authors referred to blue circles in the photo. The authors must add scale bars to the photographic images of the cross-section before and after etching process.

Corrected, those are blue circles.

  1. The reference to Figure 7 made on page 7 (line 260) should be changed to Figure 7a.

Corrected.

  1. The authors successfully demonstrate the output signal spectra in Figure 8a. How does the output signal vary with time? The authors should provide the stability data for the output signal as a function of time.

Unfortunately, the combiner burned when we investigated its pump power handling capabilities. However, during its investigation (measuring M2 factor, spectrum and polarization extinction) that took about 1 hour, we saw only 1 % power deviation at 9.1 W signal power level. – added at 298.

Please note that 10 days, given for correction of the current paper, is too short time for rebuilding and testing of the new pump combiners, so we could not add picture, which the reviewer is asking for.

  1. As shown in Figure 4, the fiber setup is placed horizontally straight for measurements. How would the output signal change if the fiber is tested under bending conditions? The authors should provide data to support the robustness of the fiber system.

The device should be packaged into a rigid housing, in which it has to be fixed in the same manner as it shown at Figure 4. It was shown outside the housing just for clarity. – added at 235.

  1. How does the output peak power depend on the diameter of the all-fiber side coupled combiner? What is the reason behind choosing specific size of 40 µm diameter for Yb-doped polarization maintained tapered fiber?

The output peak power is proportional to the core diameter of a fiber used in such a device or any other, the 40-µm core tapered fiber was used because it is a standard one our group produces – added at 213.

Round 2

Reviewer 1 Report

The authors had addressed all my questions and thus, I suggest the paper can published based on this version.

Reviewer 3 Report

The manuscript contains interesting results, and can be accepted for publication in journal Photonics in a present form.

Reviewer 4 Report

The authors have adequately addressed the comments of the reviewers. Hence, I would recommend the article to be published in Photonics in the present form.